# Effect of Thickness on Tribological Behavior of Hydrogen Free Diamond-like Carbon Coating

**Biao Huang, Qiong Zhou * and Er-geng Zhang ***

Shanghai Engineering Research Center of Physical Vapor Deposition (PVD) Superhard Coating and Equipment, Shanghai Institute of Technology, Shanghai 201418, China; sit.bhuang@163.com
* Correspondence: zhouqiong@sit.edu.cn (Q.Z.); zhangeg@yeah.net (E.Z.)

**Abstract:** The effects of film thickness on the tribological behavior have been investigated for hydrogen-free diamond-like carbon coating in this paper. The film was deposited on cemented carbide substrate (YG10C) by applying a high power impulse magnetron sputtering (HiPIMS) technique. The reciprocating ball on the disc test was conducted on the film with different thicknesses from 0.66~1.26 μm against the $ZrO_2$ ball. The friction coefficient and wear resistance of the coating with different thickness showed a unimodal change. Numerous defects were observed on the surface of the film with a thickness of 0.66 μm and the wear mechanism was mainly plow-grinding. Therefore, the steady-state friction coefficient reached the maximum value of 0.22. The coating with a thickness of 1.01 μm had a higher $sp^3$ content and a smoother, dense surface. A graphite transfer layer with low shear strength was detected on the $ZrO_2$ ball against the film with a thickness of 1.01 μm, which led to the reduction in friction, thus the steady-state friction coefficient reached the minimum value of 0.10. However, the internal stress of the film increased with increasing thickness due to the distortion of the bond angle of internal structure when the film was bombarded by high-energy particles. The peeling coating was observed under reciprocating sliding, which both played the role of plowing and boundary lubrication film. The steady-state friction coefficient was 0.14 with a coating thickness of 1.26 μm. As a result, the hydrogen-free diamond-like carbon coating with optimized thickness shows a smooth and compact surface, low internal stress, high $sp^3$ content, and better tribological properties.

**Keywords:** hydrogen-free diamond-like carbon coating; friction and wear; pulsed magnetron sputtering; coating thickness; transfer film

## 1. Introduction

Hydrogen-free diamond-like films, as a kind of DLC film (a-C film), mainly composed of $sp^2$ and $sp^3$ bonded carbon atoms have attracted great interest for their high hardness, high thermal conductivity, broad band gap, excellent anti-friction, and anti-wear performances [1–3] as well as important applications such as aerospace, biomedical, machining, and automotive parts [4–8]. However, there are some shortcomings such as high intrinsic stress, poor adhesion strength between the film and substrate, and unstable tribological properties, which limit their further development [9–13]. Most of the previous studies in recent years investigated the effect of deposition temperature [14], sputtering power [15,16], deposition pressure [16], and substrate material [17] on the tribological behavior of hydrogen-free diamond-like carbon coating. Few studies [2,18] have been found on the relationship between film thickness and tribological properties of DLC coatings, and the tribological properties of ultra-thick (more than 1 μm) DLC coating deposited by HiPIMS technique remain unclear. However, the thickness is an important factor affecting the tribological behavior of coatings, it is necessary to study the relationship between the tribological behavior of diamond-like coatings and film thickness.

Thickness has an important effect on the tribological performance of various coatings, such as TiN, CrN, and TiAlN coating [19–21], the thicker the coating, the higher the wear resistance. However, the relationship may not work for the hydrogen-free diamond-like carbon coating. The non-monotone variation relationship was presented on the growth of the coating and the degree of crosslinking of the internal carbon structure, the degree of distortion of the bond, intrinsic stress, adhesion strength, and the ability to withstand reciprocating shear force.

In this study, five groups of coatings with different thicknesses were deposited on cemented carbide (YG10C) substrates by the HiPIMS technique. The influence of the thickness on the tribological behavior was investigated. Furthermore, the effects of the thickness on the wear resistance and friction coefficient were analyzed to improve the tribological performance of the coatings.

## 2. Film Deposition

In this project, hydrogen-free diamond-like films were deposited on the cemented carbide (YG10C) substrates in a vacuum chamber with argon gas by applying the HiPIMS (PLASMAADS400) technique. The substrate is 16 mm × 16 mm × 2.5 mm in dimension with a roughness of 35 nm. The relative position between specimens and target is shown in Figure 1 and the deposition chamber is shown in Figure 2. There are 4 magnetron sputtering targets applied alternately to deposit the hydrogen-free diamond-like carbon coating, 2 Ti targets (99.9%, 49 mm in diameter) on the left and 2 graphite targets (99.9%, 49 mm in diameter) on the right. The substrates were sandblasted for 2 min with glass beads of 220 mesh under a pressure of 1.5 pa, polished by cotton wheel at a velocity of 3000 r/min for 3 min, ultrasonically cleaned with ethanol for 10 min and distilled water for 5 min, and heated in a thermostat at 120 °C for 5 min prior to deposition. The coating deposition process was then carried out by using a flow of 50 sccm Ar gas at a pressure of $1 \times 10^{-3}$ pa, the vacuum was maintained at $8 \times 10^{-2}$ pa during the sputtering process. The Ar was ionized at 2000 V and the substrates were etched for 20 min at a bias of −500 V. Ti was deposited on the substrate surface as the bottom layer to enhance the adhesion strength between the substrate and film at a repetition rate of 15 Hz for 25 min. Ti and graphite were then deposited on the bottom layer simultaneously at a repetition rate of 15 Hz for 15 min. The thickness of the film difference was altered by controlling the sputtering time of the graphite target; the sputtering time of each group varied by 20 min.

Tribological tests of the as-deposited films were performed on a ball-on-disk tribometer (UMT-3, California, CA, USA) in air and at room temperature (with the atmospheric humidity 40%). The mating ball was a $ZrO_2$ ball with 8 mm in diameter. The applied load was 10 N, the reciprocating frequency was 10 Hz with an amplitude of 10 mm. The specimens and $ZrO_2$ balls were ultrasonic cleaned for 5 min in anhydrous ethanol solution prior to the experiment. The surface morphology of specimens was analyzed by microscopy (GSX-500, Beijing, China) and the roughness was evaluated by a three-dimensional surface profilometer (Talysurf CCI Lite, London, England), the roughness test length was 8 mm and the test speed was 0.1 mm/s. The microstructures of the coating were observed by Raman spectroscopy (B&WTek bws465-532s, California, USA) with the wavelength of incident light of 532 nm. The adhesion strength was measured by the Rockwell indenter (HRS-150, Shanghai, China). The thickness of the films were measured by applying a non-destructive testing instrument (XUL-FTM, Bad Salzuflen, Germany) and the average value of the 5 sets of test data was taken as the result.

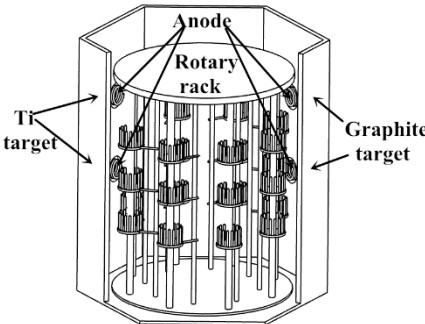

**Figure 1.** Schematic diagram of the relative position between specimen and target.

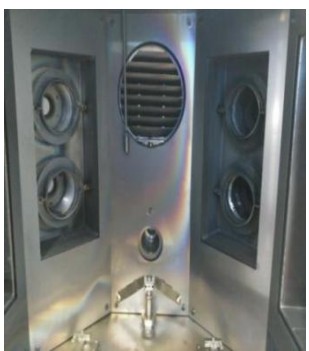

**Figure 2.** The deposition chamber.

## 3. Results and Discussions

### 3.1. Surface Morphology

The surface morphology of the films is shown in Figure 3. The surface roughness of the films is shown in Figure 4. Different micrograph and surface roughness were observed with different film thicknesses. There are deep and wide defects of large density that appeared on the coating surface (Figure 3a) and the surface roughness reaches the maximum value of 28.5 nm with a thickness of 0.66 μm. Subsequently, due to the increase of sputtering time, the defects on the film surface are found to be gradually fused and the surface roughness of as-deposited film sharply declines. When the film thickness reaches 1.01 μm (Figure 3c), the film surface clearly shows a smooth, dense micrograph with the minimum surface roughness of 9.3 nm. However, with the further increase of the film thickness, the surface performance of the film tends to decline. There are more defects observed on the coating surface with a thickness of 1.15 μm (Figure 3d) and a surface roughness of 13.1 nm, and then the surface roughness slowly increased to the value of 15.4 nm at a thickness of 1.26 μm. Such changes in surface roughness could be attributed to the reasons as follows: the surface roughness of the film is mainly affected by the surface roughness of the substrate when the film thickness is less than 1.01 μm, otherwise, the surface roughness of the film mainly depends on the value of $sp^2/sp^3$ [22–24].

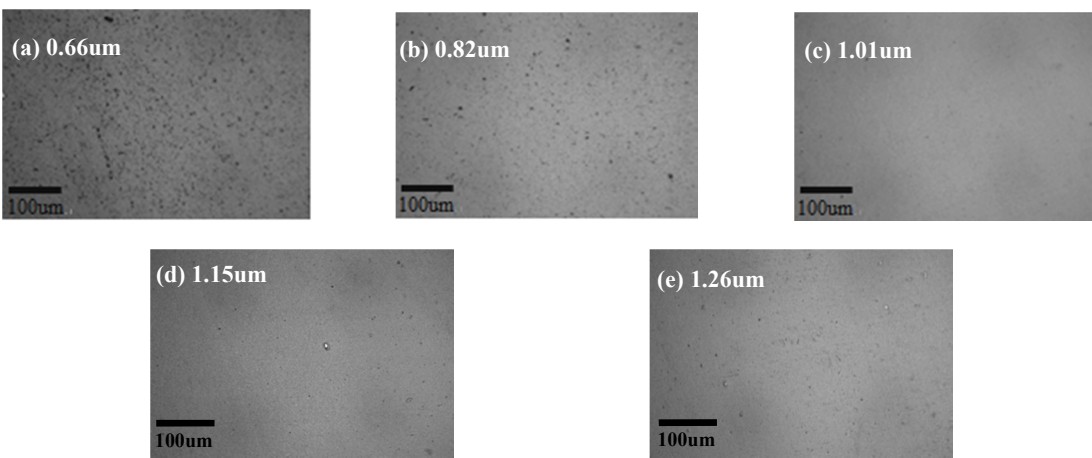

**Figure 3.** Micrograph of the hydrogen-free diamond-like carbon coating with different film thicknesses. (**a**) 0.66 um; (**b**) 0.82 um; (**c**) 1.01 um; (**d**) 1.15 um; (**e**) 1.26 um.

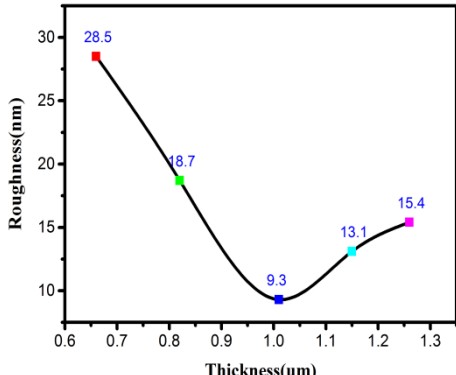

**Figure 4.** Roughness value of coatings with different thicknesses.

## 3.2. Microstructures of the Films

Usually, the Raman spectra of the hydrogen-free diamond-like carbon film are composed of a G peak at ~1560 cm$^{-1}$ and a D peak at ~1380 cm$^{-1}$. The D peak is attributed to the breathing mode of sp$^2$ bonded carbon atoms and its intensity is strongly related to the presence of six-fold aromatic rings, whereas G peak is assigned to the bond stretching of all pairs of sp$^2$ atoms in chains and rings [2,10]. Figure 5 shows a series of Raman spectra of films with different thicknesses of 0.66~1.26 μm. It is evident that the G peak is first redshifted and then blueshifted. In the thickness interval of 0.66~1.01 μm, it shows a decrease of the ID/IG from 0.693 to 0.244 and a shift of the G peak from 1564 cm$^{-1}$ to 1558 cm$^{-1}$. It also showed an increase in the number of sp$^3$ content in the film, the formation of sp$^3$ is considered to be promoted by the denser film and the stronger internal crosslinking in the process of film growth. While in the thickness interval of 1.01~1.26 μm, the G peak is blueshifted from 1558 cm$^{-1}$ to 1567 cm$^{-1}$ and the value of ID/IG increases from 0.244 to 0.747. The film was bombarded by high-energy particles accelerated by the electromagnetic field after sputtering from the target material and loses its original stress state, which leads to a decrease in the number of sp$^3$ content in the film. Therefore, the G peak locates far away from the D peak. The results are also consistent with the research of Casiraghi et al. [25] and Ferrari and Robertson [26].

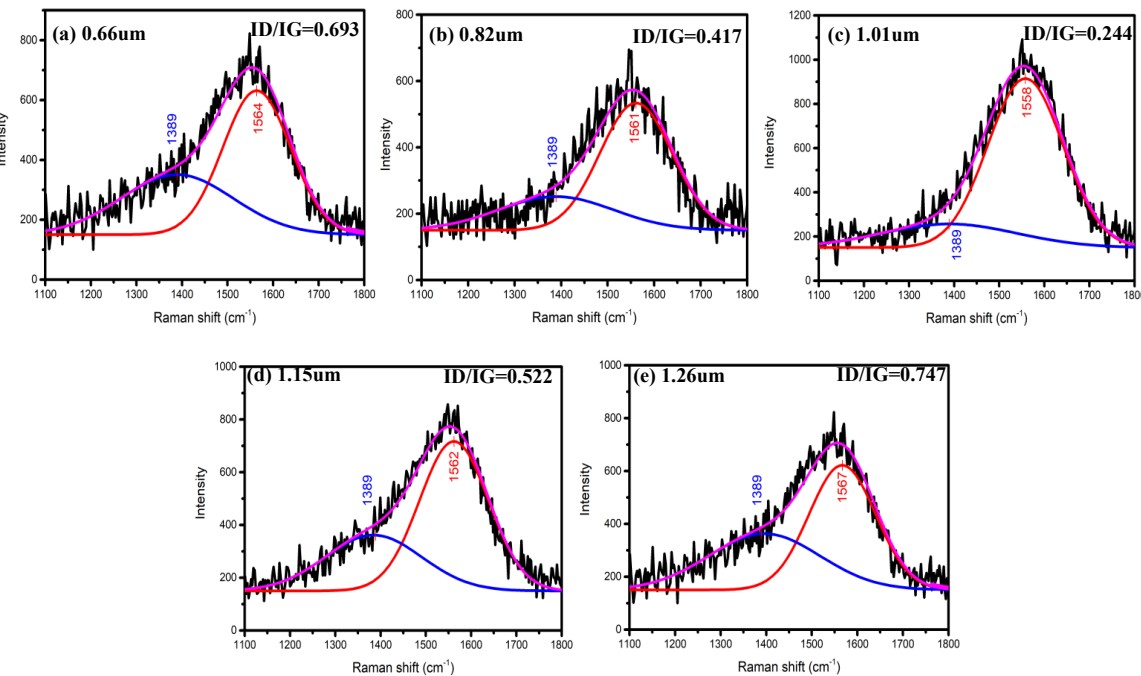

**Figure 5.** Raman spectra of deposited films with different thickness (**a**) 0.66 μm, (**b**) 0.82 μm, (**c**) 1.01 μm, (**d**) 1.15 μm, (**d**) 1.26 μm.

### 3.3. Adhesion Strength

The adhesion strength of the as-deposited film was shown in Figure 6. According to the VDI3198 standard from the German Association for Science and Technology, the loading force is 60 N and the specimen is observed under a 100× microscope. The adhesion strength of the film with a thickness of 0.66~0.82 μm is HF3. The adhesion strength of the films with a thickness of 1.01 μm is HF1 is the best. While the adhesion strength of the films with a thickness of 1.15 μm and 1.26 μm are HF3 and HF5, respectively. The indentation results showed first an increased adhesion strength and then a decreased adhesion strength of the films with a film thickness increasing from 0.66 μm to 1.26 μm. The film with a thickness of 0.66 μm showed many defects, a poor uniformity, and low external force bearing capacity. The film was subjected to uneven stress under the effect of external pressure, the substrate was more prone to deformation and in this case, the coating was easy to peel off from the substrate. With the increase of the thickness, the films showed a stronger bearing capacity, a denser and more uniform film layer, and lower intrinsic stress. In that case, the film was subjected to even stress under the effect of external pressure, only small micro-cracks were formed. However, with a further increase in the thickness, the bond angle was distorted by the bombardment of high-energy particles, which resulted in a sharp rise in the intrinsic stress, thus the coating became more brittle. When the film was subjected to external pressure, the deformation of the film and the substrate were not in step with each other, therefore, the film peeled off from the substrate into pieces and exhibited very low adhesion strength.

### 3.4. Tribological Properties

The relationship between the coefficient of friction (COF) and the film thickness was shown in Figure 7. It is obvious that the thickness strongly affected the friction coefficient of the films. The films with different thicknesses all went into the stable wear stage within 7200 cycles after the running-in stage. It is worth mentioning that after the running-in-period, the COF value of the film with a thickness of 0.66 μm remains 0.15 around 1600 cycles and then increases slowly to 0.22. Tiny peaks with slow slope change appeared on the friction coefficient curves with a thickness of 0.66 μm and 0.82 μm, which was considered to be caused by changes in the shape of surface defects during the friction and wear process. The film with a thickness of 1.01 μm had smooth wear and a minimum

friction coefficient of 0.1. The friction coefficient slowly rising to 0.12 with the thickness increased from 1.01 μm to 1.15 μm and continually rising to 0.14 when the thickness reached up to 1.26 μm. A peak span of about 200 cycles appeared on the friction curve of the film with a thickness of 1.26 μm.

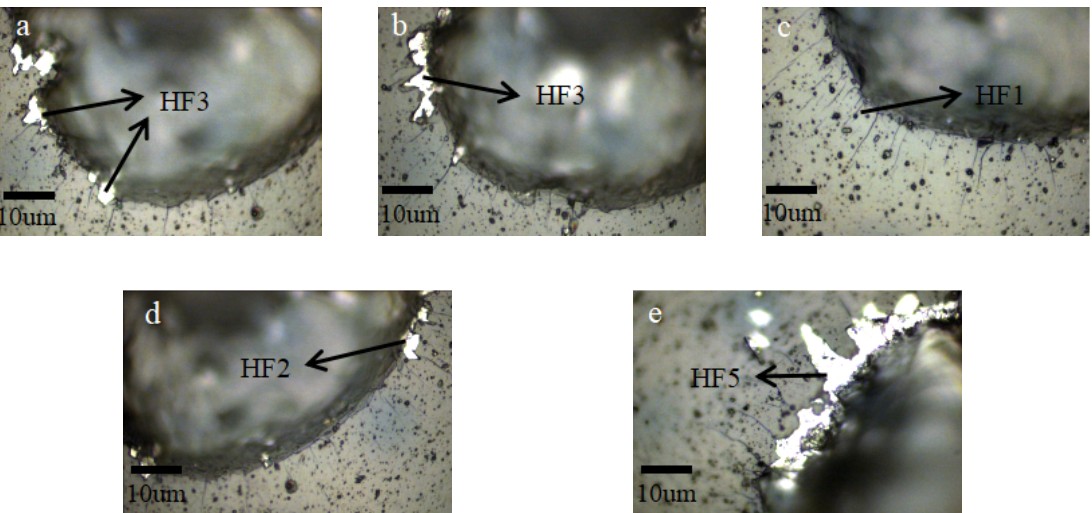

**Figure 6.** Surface indentation of the films with different thicknesses under a microscope. (**a**) 0.66 μm, (**b**) 0.82 μm, (**c**) 1.01 μm, (**d**) 1.15 μm, (**e**) 1.26 μm.

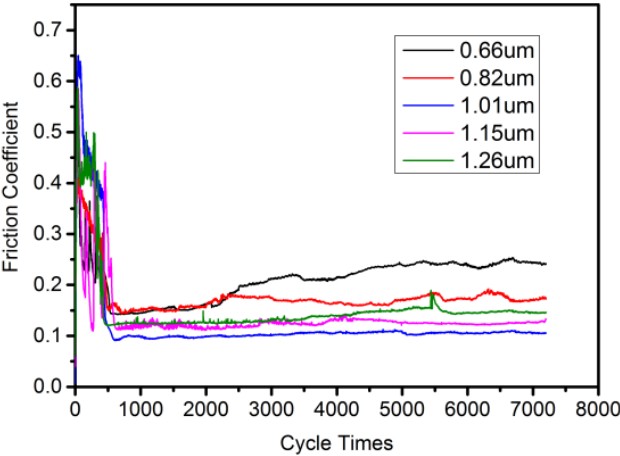

**Figure 7.** The relationship between the coefficient of friction and the thickness.

　　The wear morphology of the films with various thicknesses were shown in Figure 8. The wear resistance of the as-prepared film exhibited a trend of first increasing and then decreasing, with the film thickness increasing from 0.66 μm to 1.26 μm. There is a disturbance of about 0.1−0.2 nm on the curve of Figure 8 which may be attributed to the contact probe of the applied equipment vibrates when it encounters micro-asperities. However, this effect can be ignored for the characterization results. In the thickness interval of 0.66~1.01 μm, the wear of the film gradually decreased, and the abrasion resistance was on the rise. In the thickness interval of 1.01~1.26 μm, the wear of the film gradually increased, the abrasion resistance gradually reduced, and two peaks were observed in the deep ground of the wear track. The film with a thickness of 0.66 μm (Figure 8a) was worn out with a wide and deep wear track. That rapid wear was considered to be caused by the high surface roughness of the film, which led to the large shear force at the asperities due to the uneven stress. The film with the thickness 1.01 μm (Figure 8c) obtained the minimum wear track width. As the film worked under more uniform stress, which had little effect on the variation of the shear force. The sp$^3$ content of the film was relatively high, as well as the density of the σ bond with strong bonding C-C type which was

not easily broken down by the friction [2]. As the thickness increased to 1.26 μm (Figure 8e), a deeper wear track was found with two peaks in the deep ground. The scratched area was clearly separated from the unscratched area, thus it could be concluded that the coating has been peeled off.

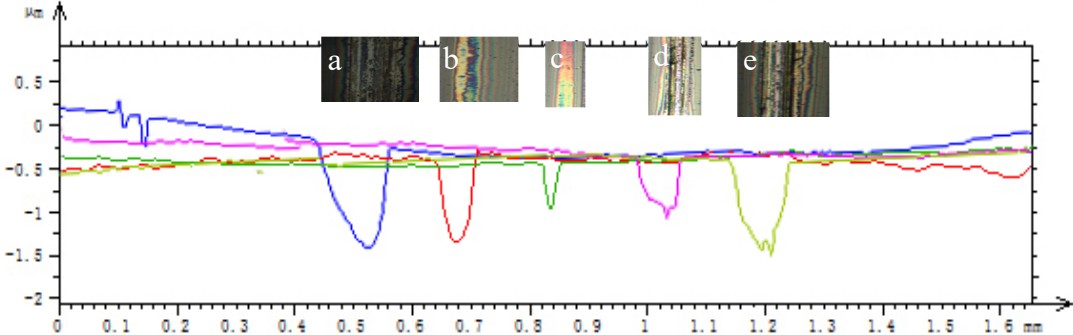

**Figure 8.** The wear morphology of the film with various thicknesses: (**a**) 0.66 μm, (**b**) 0.82 μm, (**c**) 1.01 μm, (**d**) 1.15 μm, (**e**) 1.26 μm.

To investigate whether there is a transfer on the counter-body surface. The grinding surface of the $ZrO_2$ ball was evaluated using Raman spectra after the friction test, as shown in Figure 9. The signal only appears in the counter-body against the film with thickness 1.01 μm, a peak at ~1560cm$^{-1}$ became evident and sharp, which can be identified to the G peak. The D peak can be identified as the small shoulder peak at ~1390cm$^{-1}$. The strength and area of the G peak are much larger than that of the D peak, the Raman spectra also reveal a typical structure of the graphite phase [2,10]. The graphite layers would slide over one another during the friction process, which has a good lubrication effect. This may be the reason for the lowest friction coefficient of the film with a thickness of 1.01 μm.

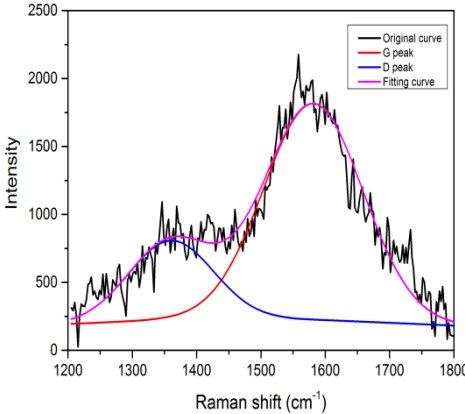

**Figure 9.** Raman spectra of the grinding surface of the $ZrO_2$ ball and the film with a thickness of 1.01 μm.

Through the above findings, it is evident that the film thickness has a significant influence on the tribological properties of the hydrogen-free diamond-like carbon films. When the film thickness is below 0.66 μm, the film surface had numerous defects that led to the severe plowing effect in the friction process against the $ZrO_2$ balls, and this plowing effect could not be relieved because the wear debris captured by the pinholes were unable to fill the pinholes. Moreover, the convex defects on the film surface made the film worked under uneven force and the convex structure was subject to a larger shear force that was worn rapidly, which results in the relatively high friction coefficient and poor wear resistance of the film. However, the hydrogen-free diamond-like carbon films had $sp^2$ structure, the sideslip of which would reduce the friction coefficient. Therefore, although there was a severe

plowing effect on the film, the friction coefficient of the hydrogen-free diamond-like carbon coating was still lower than that of the TiNx, TiAl$_{1-x}$N$_x$, and nc-TiAl$_{1-x}$N$_x$/SiN$_x$ coatings [27].

With the increase of film thickness, the defects gradually integrated. The film gained a smooth and dense surface with a thickness of 1.01 μm, the wear debris captured by the pinholes were able to fill the pinholes which could relieve the plowing effect. The graphite transfer film formed on the ZrO$_2$ ball surface which can reduce the plowing effect and work as a lubricant. Thus, the friction coefficient showed a sharp decline and got a stable value of 0.10. In addition, the high sp$^3$ content could increase the wear resistance of the films as well. As the film thickness continues increasing, the film was bombarded by high-energy particles, and the bond angle was distorted, which results in a sharp rise in intrinsic stress. When the film was subjected to external pressure, it peeled off into pieces, thus the film performed low wear resistance and showed deep wear track. The coating pieces that peeled off would scratch the surface that was working in the following reciprocating motion under normal pressure, and two peaks appeared in the deep ground of the wear track. Whereas graphitization occurred on parts of the coating pieces peeled off at an elevated temperature, and the friction coefficient slowly raised to 0.14 under that boundary lubrication.

## 4. Conclusions

(1) The surface morphology, sp$^3$ bond content and adhesion strength of the hydrogen-free diamond-like carbon film first increased and then decreased by increasing the film thickness from 0.66 μm to 1.26 μm. The film with a thickness of 1.01 μm had the best performance with smooth and dense surfaces and the highest sp$^3$ bond content and the adhesion strength reached up to HF1.

(2) When the film thickness is too thin, the island structure on the surface was not fused, thus the plowing effect was obvious in the friction process. The convex island structure on the surface made the film work under an uneven force, which resulted in a high friction coefficient and poor wear resistance. While if the film thickness is too thick, the internal structure bond angle was seriously distorted with high intrinsic stress, therefore, the film was prone to peel off the substrate.

(3) During the wear process of the as-deposited film against the ZrO$_2$ ball, graphitization was generated on the wear debris due to high contact temperature and high contact stress. The transfer film was formed on the ZrO$_2$ ball surface, which had a lubrication effect on the friction area. As a result, the friction coefficient was reduced.

**Author Contributions:** B.H. and Q.Z. conceived, designed and performed the experiments; B.H. and E.-g.Z. analyzed the date and wrote the paper; B.H., Q.Z. and E.-g.Z. provided materials/analysis tools. All authors have read and agreed to the published version of the manuscript.

**Funding:** This research was funded by the Natural Science Foundation of China (grant number 51971148) and Key Support Plan of Shanghai Science and Technology Committee (grant number 170905038000).

**Conflicts of Interest:** The authors declare no conflict of interest.

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
