# Peer review of "Effect of Thickness on Tribological Behavior of Hydrogen Free Diamond-like Carbon Coating"

_coatings, doi:10.3390/coatings10030243_

Round 1

Reviewer 1 Report

A good paper.  However, you present an hypothesis on the impact of thickness on the properties, and discuss this, but do not suggest further ways of testing that hypothesis and so justifying your conclusions. 

1.What experiments or theoretical approach do you propose in order to complete the validation of your conclusions?  This needs to be addressed at least in principle.

Author Response

Dear Reviewer

    Thank you very much for your suggestions. However, in this paper, the effect of thickness on film properties has been verified by various experiments such as microscope and Raman test, tribological test. Results shows that the thickness do have significant effect on the surface quality, internal structure, internal stress and tribological performance (friction coefficient and wear resistance), which can justify our conclusions.

Reviewer 2 Report

The effects of film thickness on the tribological behavior have been investigated for hydrogen free diamond-like carbon coating in this manuscipt. The hydrogen free diamond-like carbon film was deposited on cemented carbide substrate (YG10C) by applying high power impulse magnetron sputtering (HiPIMS) technique. I find the topics of the manuscript is interesting, but I have several comments which should be answered before consideration for publication:

1.In Figure 3. scales and "d" "e" fragments etc. should be corrected. Captions a, b, c should be placed on microscopic images. Details of the thickness should be in the signature under the figure.

2.It could be useful to attach SEM images showing defects on the substrates.

3.An AFM measurement could be useful to assess surface roughness.

4.I would also like to ask for more details about the deconvolution of Raman spectra. (was peak D fixed?)

5.It is difficult to assess the shift of the G peak at such a noisy spectrum. When assessing peak shifts, you need to use a higher quality Raman spectrum to effectively evaluate the location of the D and G peaks.

6.The graph frame obscures the ID / IG calculations.

7.The position of the ID peak on the spectrogram is obstructed by the spectrum.

8.Arrange Figure 6 similarly to Figure 5. Signatures (e.g. (d) 1.15 um) should be included in the description of the figure 6.

9.Chart 7 should be extended with a COF dependency (e.g. for 1000 cycles) on the layer thickness.

10.chart 7 should be extended with COF dependence (e.g. for 1000 cycles) on the roughness of the layer.

11.figure 8 no markings a, b, c ...

12.what are the disturbances about 0.1-0.2 nm on figure 8?

Author Response

Dear Reviewer

  Thank you very much for your suggestions. I’m sorry that China has "coronavirus" outbreak at present. And the laboratory equipment is unavailable, so above tests cannot be carried out recently. Although the surface morphology observed under an optical microscope in Fig. 3 is not as clear as that is obtained by SEM, Fig.a, b, c, d and e in Fig. 3 are all obtained by using the same instrument, so Fig.3 can be used to analyze the differences of film surface quality with different thicknesses, as well as the surface roughness of the film and Raman specturm.

I would also like to ask for more details about the deconvolution of Raman spectra. (was peak D fixed?)

D peak is not fixed, I think it may be a coincidence that the peak position of D peek is the same for the 5 groups of data.

what are the disturbances about 0.1-0.2 nm on figure 8?

The contact probe of the applied equipment vibrates when it encounters micro-asperities, which leads to the disturbances. However, this effect can be ignored for the characterization results.

Reviewer 3 Report

The author studied the effect of film thickness on the tribological behavior of hydrogen free diamond-like carbon coating. Among five thicknesses the layer with 1.01 um showed the best structural and functional properties. The research idea, procedure, and results and discussion are very understandable and easy to follow. However, I have some points for author consideration:

I highly recommend abbreviating “hydrogen free diamond-like carbon” term as mentioned 27 times through the manuscript. In the abstract, lines 16-18, it mentioned that the graphite was detected ZrO2 ball without specifying to which thickness. However, page 6, line 188 states “The signal only appears in the counter-body against the film with thickness 1.01um”. explain, please. In the introduction, I think an additional few sentences about hydrogen free diamond-like carbon structure will make the paper more understandable. Page 2, line 67-69 in the film deposition section, it is reasonable to add Ti to improve the adhesion. However, it is not very clear why Ti and graphite were deposited on the bottom layer simultaneously, explain, please. Please remove ‘a’ and ‘b’ from fig 1&2 In fig 3, do you have images with higher magnification to show the defects? How you confirmed those dark areas in the film are defects? Have you done any EDS mapping to them? Figure 8 also the micrographs not very clear and missing the sale bar. In addition, the graph also has no legend. If I compare Raman spectra for coating with 1.01um (fig 5 c) and the Raman for the same coating, after deposition on the ball, fig 9, I think ID/IG ratio for the first figure much lower than the second one. Is the graphite quality changed? Explain, please.

Author Response

Dear Reviewer

   Thank you very much for your suggestions. My reversion as follows:

In the abstract, lines 16-18, it mentioned that the graphite was detected ZrO2 ball without specifying to which thickness. However, page 6, line 188 states “The signal only appears in the counter-body against the film with thickness 1.01um”. explain, please.

Maybe it is not clearly stated in the abstract, the results of the experiment showed that the transfer film was only observed on the surface of ZrO2 ball anti the film with the thickness 1.01um.

Page 2, line 67-69 in the film deposition section, it is reasonable to add Ti to improve the adhesion. However, it is not very clear why Ti and graphite were deposited on the bottom layer simultaneously, explain, please.

The layer of titanium and carbon were deposited on the bottom layer simultaneously in order to reduce the influence of thermal expansion coefficient difference between the titanium layer and the hydrogen free diamond-like carbon layer on the film performance, and improve the stability of the film.

Please remove ‘a’ and ‘b’ from fig 1&2 In fig 3, do you have images with higher magnification to show the defects? How you confirmed those dark areas in the film are defects? Have you done any EDS mapping to them?

I'm sure the dark area is defect, because the specimens used for the test have been ultrasonic cleaned clearly by using anhydrous ethanol and the pollutants on the surface have been removed. I’m sorry that China has "coronavirus" outbreak at present. And the laboratory equipment is unavailable, so EDS tests cannot be carried out recently.

 Figure 8 also the micrographs not very clear and missing the sale bar. In addition, the graph also has no legend. If I compare Raman spectra for coating with 1.01um (fig 5 c) and the Raman for the same coating, after deposition on the ball, fig 9, I think ID/IG ratio for the first figure much lower than the second one. Is the graphite quality changed? Explain, please.

The Raman test results in Fig. 5 show the internal structure of the film on the specimens, while the Raman test results in Fig. 9 show the powdery "condensed" carbon layer mechanism, which are essentially different from each other. The hydrogen free diamond-like carbon film is extruded and transferred to the surface of ZrO2 ball in powder form. This process is accompanied by the graphitization of the film, therefore the structure of the transfer film also changed.

Round 2

Reviewer 2 Report

I fully understand the Chinese problem with "coronovirus". But the changes made are not enough. I suggest asking the editor to extend the response time until the laboratory is fully opened. Answers to questions asked in the previous version of the review are still required. Additionally:

1.all the time figure 3 needs improvement (Figure (d) and Figure (e) - signatures).

2.still in figure 5 the value of peak D (1389cm-1) is not visible.

3.in figure 6 it is required to add layer thickness in the description under figure.

"The contact probe of the applied equipment vibrates when it encounters micro-asperities, which leads to the disturbances. However, this effect can be ignored for the characterization results." This explanatory comment would be required in the manuscript text.

Author Response

Dear Reviewer

     Thank you very much for your suggestions. I have modified it in the revised version.
